# Imaging of the Liver and Pancreas: The Added Value of MRI

**DOI:** 10.3390/diagnostics14070693

**Published:** 2024-03-26

**Authors:** Giovanni Morana, Alessandro Beleù, Luca Geraci, Luisa Tomaiuolo, Silvia Venturini

**Affiliations:** Radiological Department, General Hospital Treviso, 31100 Treviso, Italy; alessandro.beleu@aulss2.veneto.it (A.B.); luca.geraci@aulss2.veneto.it (L.G.); luisa.tomaiuolo@aulss2.veneto.it (L.T.);

**Keywords:** liver, pancreas, magnetic resonance, contrast media, secretin

## Abstract

MR is a powerful diagnostic tool in the diagnosis and management of most hepatic and pancreatic diseases. Thanks to its multiple sequences, the use of dedicated contrast media and special techniques, it allows a multiparametric approach able to provide both morphological and functional information for many pathological conditions. The knowledge of correct technique is fundamental in order to obtain a correct diagnosis. In this paper, different MR sequences will be illustrated in the evaluation of liver and pancreatic diseases, especially those sequences which provide information not otherwise obtainable with other imaging techniques. Practical MR protocols with the most common indications of MR in the study of the liver and pancreas are provided.

## 1. Introduction

To assess the presence of metastases in the liver, imaging of the liver and pancreas takes the lead in the management of most diseases affecting the abdomen and also in the management of oncological patients.

Although US and CT are the most used imaging techniques worldwide, MRI is assuming an increasingly important role due to specific imaging features that can offer information not otherwise obtainable.

Although contrastographic imaging can be obtained with US (CEUS), CT (CECT) and also with MRI, each method offers different sensitivities depending on the contrast agent (CA) used, and it is precisely MRI, thanks to its high sensitivity to CAs, that offers the greatest accuracy in identifying the enhancement of lesions, thanks also to the development of CAs with increasingly high relaxivity that have entered the market in recent months (e.g., Gadopiclenol, Bracco and Guerbet) [1].

However, the specificity of MRI goes beyond contrastographic imaging, which, although of greater sensitivity and specificity, is essentially superimposed on that of the other methods.

On the other hand, other information can be obtained with MRI that cannot be obtained with other methods, and that represents the uniqueness of the method, such as guaranteeing an information capacity that is clearly superior to other methods, capable of offering sensitivity and specificity values that are clearly superior to CT and US.

This article will address the specific imaging modalities of MRI distinct in the liver and pancreas.

## 2. Liver

The liver is involved in various body functions and in supporting other organs. Due to the rising prevalence of hepatic diseases, imaging studies are increasingly utilized for its evaluation. US is a widely available technique usually used as a first approach, while CT is mostly used as a second level technique in case of indeterminate focal lesions with US, and as the first imaging tool in the case of staging and follow-up of oncological patients.

MRI, with its high contrast resolution, non-morphological sequences and the use of liver-specific contrast agents, is a key tool for the comprehensive evaluation of liver diseases, especially in the detection of small lesions, characterization of atypical lesions, quantification of iron and fat accumulation, as well as fibrosis; although, its use is case-dependent due to higher costs and limited availability.

The most specific and useful MR sequences which can offer an added value to the analysis of the liver are:

**a. Gradient multi-echo sequences**: The lack of a 180° refocusing pulse in this sequence determines susceptibility and chemical shift artifacts that help to identify and quantify fat and iron deposition within the liver parenchyma to better depict diffuse liver disease and focal liver lesions.

- **Fatty deposition**: The GRE in-phase and out-of-phase MR sequence is a T1 sequence with a dual echo acquisition in fixed temporal intervals (the second one double the first one) which reveals chemical shift artifacts, a result of fat protons having a lower precession frequency than water protons. In the in-phase state, signals constructively sum, displaying combined water and fat signals. Conversely, the out-of-phase state exhibits signal opposition related to a 180° phase difference between water and fat protons, thus nullifying water and fat signals. This sequence aids intracellular fat detection in hepatic lesions or parenchyma. Notably, liver conditions like steatosis manifest significant fat content, visible in out-of-phase images as hypointensity due to signal cancellation. Moreover, some hepatic lesions such as adenoma or HCC may show intracellular fat deposition, again visible in out-of-phase images as a drop of signal intensity due to the cancellation of signal related to the opposition of water and fat in the same voxel (Figure 1 and Figure 2). In-phase and out-of-phase MR sequences should always be included for a complete assessment of the liver [2].

- **Iron quantification**: Iron overload is a systemic disorder with high iron serum levels and increased iron storage in the form of ferritin and hemosiderin. The liver is the first organ that shows iron overload because one of its functions is to store iron within hepatocytes and Kupffer cells. Calculating the amount of iron deposition can lead to a better management of patients that suffer from this disease as an untreated high iron overload may lead to a cirrhosis with a risk to develop an HCC 20-fold higher than in the general population [3].

Tissue accumulation of iron creates local magnetic field inhomogeneity which causes the transverse magnetization (T2) to decay much faster than would be predicted, with an effective T2 value denoted as T2*. To quantify the amount of iron deposition, the most common technique is to acquire a breath-hold gradient-echo sequence with progressively increasing echo times (ETs), specific for each strength of magnetic field (1.5T, 3T). The extracted signal intensity curve is fitted, resulting in an estimate of liver T2* which is proportional to the amount of iron deposition (Figure 3).

***b. GRE T1-weighted 3D sequence*:** This sequence is utilized in the dynamic imaging after the injection of a gadolinium-based contrast agent; high-contrast levels after the bolus and thin slices make it possible to reconstruct the vessels with MIP or VR techniques. Although this information is similar to what is obtained with a contrast-enhanced CT, it is possible to obtain multiple arterial acquisitions (generally 2–3), with a multiphasic contrast-enhanced MRI sequence that allows for the acquisition of multiple arterial subphases within a single breath-hold. This sequence facilitates the timing of the arterial phase and adds dynamic characteristics of focal lesion vascularization [4]. (Figure 4).

**c. Biliary imaging**: The visualization of biliary ducts with MRI is easily obtained with a heavily T2-weighted sequence, 2D or 3D, using the water in the fluid as an intrinsic contrast agent [5]. In 3D, acquisition images are then reformatted in different planes using maximum intensity projection (MIP).

The long T2 relaxation time of the water causes the surrounding tissues to be markedly hypointense during the acquisition of the images. With this technique, many biliary pathologies can be explored, such as congenital biliary anomalies, biliary lithiasis, jaundice, sclerosing cholangitis, central cholangiocarcinoma, etc., with a delineation of biliary ducts far superior than that of CT (Figure 5). 

Moreover, by using liver-specific MR contrast agents during the biliary phase, with a high-resolution 3D T1 sequence and MIP reformation, it is possible to obtain a functional visualization of the choledochus and biliary ducts up to the second order. With this technique, it is possible to assess the functionality of biliary anastomosis, the passage of the bile in the duodenum, which can be impaired in the case of sphincter of Oddi dysfunction (SOD) or to confirm a bile leak, either after trauma or surgery [6,7,8] (Figure 6).

**d. Diffusion Weighted Imaging** (DWI) is a technique which investigates the water content of organs and tissues, establishing the random motion of water molecules in a single voxel called Brownian motion. This technique is able to distinguish water protons free to diffuse in a tissue from others which are not, and the mechanism is due to a different microscopic spread of water molecules. In vivo, tissue structures such as cell membranes prevent the motion of water molecules through the interstitial space; then, high-cellularity tissues can be sorted from those with lower packed cells, or from free fluid. Moreover, according to the molecular structure in which water protons are embedded, the diffusion of water protons can be different; proteins, blood and other structures impair the diffusion of water molecules, thus showing a restricted diffusion [9].

Technically, a DWI sequence is based on the application of two temporally spaced gradient pulses with the same strength but opposite direction; water protons which freely diffuse in the tissue lose the signal and thus are not visible in the resulting image, while water protons with restricted diffusion still maintain the signal and appear hyperintense in the resulting image. The intensity of each voxel’s image element reflects the estimated rate of water diffusion at that location.

The level of water restriction can be quantified by means of the Apparent Diffusion Coefficient (ADC) map, obtained with at least two different b values in a DWI sequence, which is a parameter of the sequence, whose value starts at 0 and can be widened up to 2000 (depending on the external magnetic field), modifying the gradient amplitude and duration by widening the interval between paired gradient pulses.

DWI has been proposed as a useful tool in several conditions [10]:

- **Identification** of focal liver lesions: DWI with a b value < 100 shows the highest sensitivity in the identification of focal liver lesions [10] (Figure 7). DWI has the best sensitivity to detect liver metastases, especially small lesions. A combination of DWI and contrast-enhanced T1w images shows the best performance in the detection of liver lesions compared to each sequence alone [11]. In patients who cannot receive a gadolinium-based contrast agent (GBCA), DWI is a reasonable alternative.

- **Characterization** of focal liver lesions: The ADC map is useful to distinguish lesions with restricted diffusion, such as malignant lesions (e.g., metastases) (Figure 7), from lesion without restricted diffusion, such as benign lesions (e.g., hemangiomas) [10] (Figure 8), but up to now DWI with ADC quantification cannot reliably discern between solid benign and malignant lesions or between different malignant lesions. Although DWI is not very helpful in the detection of small HCCs, it can be useful in characterizing small atypical nodules in a cirrhotic liver, whose restricted diffusion can be a predictor of a premalignant condition [12]. DWI with ADC quantification, moreover, is highly helpful in distinguishing necrotic lesions (Figure 9) from abscesses (Figure 10), whose morphological appearances can be indistinguishable [13].

- **Assessment** of therapeutic response: DWI is a promising non-invasive tool for assessing therapy response in liver metastases [14] and HCC [15]. Changes in ADC values are related to tumor necrosis and anticipate changes in the size or enhancement of lesions (Figure 11). Moreover, some studies suggest that initial ADC values can be a predictor of treatment response, although further studies are necessary to validate these results [16].

- Evaluation of diffuse liver disease: although different studies have attempted to correlate non-alcoholic fatty liver disease (NAFLD), fibrosis and cirrhosis to DWI parameters, no definitive or clear results have been proved; thus, at present, DWI cannot be used as a biomarker for diffuse liver disease [17].

**e. T1 and T2 mapping**: T1/T2 mapping are parametric maps which exploit the longitudinal (T1) or transversal (T2) relaxation time, i.e., the time required for longitudinal magnetization to return to equilibrium after an inversion or saturation pulse (T1) and the time required for the loss of phase coherence of transverse magnetization after an excitation pulse (T2).

- **T1** is related to water concentration in the tissue, the level of protein concentration (higher the level, shorter the T1), iron and rough endoplasmic reticulum. Moreover, the presence of GBCA greatly influences the T1: after I.V. injection, GBCA distributes in the interstitial and intravascular space, reducing tissue T1. Thus, T1 mapping in unenhanced images can provide information about tissue composition, such as water, collagen, protein, lipid and even iron content. T1 mapping acquired before and after GBCA injection can provide information on extracellular volume (ECV), an index of fibrosis and edema [18]. The extracellular volume fraction (ECV) represents the extracellular compartment. ECV is strongly related with the extracellular matrix and can be used as an important diagnostic biomarker for fibrosis and edema [19]. Moreover, T1 mapping can be acquired after the injection of liver-specific MR contrast agents, such as Gadobenate Dimeglumine (Multihance, Bracco, Milano) and Gadoxetate disodium (Primovist, Bayer, Berlin) during the hepatobiliary phase of excretion. Furthermore, several studies have found a correlation between epatobiliary contrast-enhanced T1 mapping and ECV with the histological amount of hepatic fibrosis, liver function tests and Child–Pugh scores [20].

- **T2** mapping generates a parametric map which is related to water concentration and the presence of superparamagnetic substances such as iron and deoxyhemoglobin, which generate local field heterogeneity, thus reducing tissue T2. 

**f. Liver-specific MR contrast agents** (LSCAs): these compounds are injected via I.V. after an initial distribution in the vascular-interstitial compartment; similar to other GBCAs, part of the injected dose is collected by hepatocytes, with an enhancement of the liver parenchyma due to the paramagnetic effect. 

Two molecules belong to this group. 

- **Gadobenate dimeglumine** (Gd-BOPTA, Multihance^®^, Bracco, Milan, Italy) is a gadolinium chelate with a slight and transient protein bonding and an elimination profile with approximately 96 per cent of the injected dose eliminated via glomerular filtration, while the remaining 2–4 per cent is taken up by functioning hepatocytes and eliminated through the hepatobiliary pathway. The interaction with albumin gives Gd-BOPTA double the relaxivity than a conventional paramagnetic MDC, while the uptake by the hepatocytes leads to an increased enhancement of the liver parenchyma 1 h to 3 h after administration. 

- **Gadoxetic Acid** (Gd-EOB-DTPA, Primovist^®^, Bayer, Berlin, Germany), similarly to Gd-BOPTA, after bolus injection, is initially distributed in the vasculo-interstitial compartment; thereafter, 50 per cent of the injected dose is carried by hepatocytes through an organic anion transporter and eliminated via the biliary pathway. The increase in hepatic parenchymal enhancement is earlier, starting as early as 5 min after injection, and at 20 min optimal liver signal enhancement is obtained, which lasts about 2 h. 

While Gd-BOPTA, thanks to its high relaxivity, has a better signal during the dynamic phase, Gd-EOB-DTPA offers a better enhancement in the hepatobiliary phase thanks to its 50% elimination route via the biliary system [21].

Both LSCAs during the dynamic phase behave similarly to conventional paramagnetic MDC, while in the late phase they improve the sensitivity of MRI in identifying focal liver lesions and increase its specificity, contributing to a better characterization of lesions [22]. 

Intracellular uptake of these CMs occurs via the organic anion transport polypeptide (OATP), and then they are excreted in the bile with an ATP-dependent system. The presence of such transport systems and a normal representation of the biliary canalicular system within the lesions are necessary conditions to uptake these CMs. 

In principle, benign hepatocytic lesions such as FNH are able to pick up such MDC, thus appearing hyperintense in the hepatobiliary phase [23] (Figure 12). On the other hand, in borderline/malignant hepatocellular lesions, from hepatic adenomas to HCC, the presence of such transporter systems is very poor, so that on average these lesions appear hypointense in the hepatobiliary phase [24,25,26] (Figure 2 and Figure 13). All non-hepatocytic lesions, whether benign (e.g., angiomas) or malignant, whether primary (cholangiocarcinoma) or secondary (metastases), are unable to pick up such MDC, thus appearing hypointense in the hepatobiliary phase (Figure 14). In solid hypervascular liver lesions, a liver-specific MR contrast agent is crucial for the differential diagnosis [27].

**- Biliary imaging with positive contrast media**: with both Gd-BOPTA and Gd-EOB-DTPA, it is possible to visualize the biliary system during the biliary excretion phase. By using GRE 3D sequences, MIP images can subsequently be generated. In such a way, both a morphological and functional analysis of the biliary tract can be performed. T1-enhanced MRCP is useful for assessing the patency of a biliary anastomosis or a bile leak after surgical procedure or trauma (Figure 6).

In Table 1, different practical protocols of MR hepatic imaging in different clinical situations are offered, together with the most useful indications of MRI in different clinical situations.

## 3. Pancreas

Although Endoscopic Ultrasound (EUS) is the technique which shows the best accuracy in the detection and characterization of pancreatic diseases, thanks also to the possibility to perform fine needle biopsy, it is limited by low availability and operator dependency. For this reason, MRI plays a leading role in the imaging of the pancreas, especially with recent technical innovations such as breath hold T1- and T2-weighted images and respiratory triggered T2-weighted images, as well as dynamic imaging after injection of contrast agent and the administration of secretin. MR shows a great capability to explore pancreatic ducts, vessels and parenchyma with a non-invasive approach. The most specific and useful MR sequences which can offer an added value to the analysis of the pancreas are:

**a. Fast spin-echo (HASTE, RARE) sequence:** These are single-shot turbo spin-echo sequences with half acquisition of the K space, with an acquisition time for each slice of 1 s. The goal of this sequence is a low sensitivity to movement artifacts, which is more suitable for non-cooperative patients. A great sensitivity to fluids, which appear highly hyperintense, leads to a better visualization of the pancreatic duct and all the cystic lesions, the peripancreatic area such as stomach and duodenal content, and peripancreatic fluid collection. On the other hand, low sensitivity in the detection of small and low-contrast solid lesions is one of its disadvantages. A normal pancreas demonstrates a similar or higher signal compared to liver parenchyma, whereas biliary and pancreatic ducts appear highly hyperintense (Figure 15).

**b. GRE T1-weighted 2D sequence with fat saturation:** The normal parenchyma of the pancreas appears homogeneously hyperintense, which is related to the presence of aqueous protein in the acini, the abundance of endoplasmic reticulum within the acinar cells and the content of manganese. On unenhanced imaging of the pancreas, those features make this sequence the one which better differentiates normal from pathological parenchyma. Indeed, fibrosis and fatty infiltration reduce the high signal of the pancreas that appears hypointense. This sequence shows a great sensitivity to depicting pancreatic disease but cannot differentiate various pancreatic lesions. (Figure 16 and Figure 17).

**c. GRE T1-weighted 3D sequence**: As already discussed in the liver chapter, special multiphasic contrast-enhanced MRI allows for the acquisition of multiple arterial subphases within a single breath-hold. This sequence facilitates the timing of the arterial phase and leads to more dynamic information of focal lesion vascularization [4]. Considering pancreatic pathology, it is valuable in the detection of hypervascular tumors such as neuroendocrine ones.

**d. Diffusion Weighted Imaging (DWI):** In recent years, long acquisition time and upper abdomen physiological artifacts such as bowel peristalsis, blood flow and respiratory movements prevented the application of DWI sequences. Nowadays, parallel imaging and respiratory triggering allow for the routine application of DWI in the upper abdomen [28].

A recent meta-analysis demonstrated a sensitivity of 83%, a specificity of 87% and an AUC of 0.92 for quantitative DWI and ADC in distinguishing benign from malignant lesions [29]. However, another meta-analysis showed poor results for ADC in differentiating pancreatic adenocarcinoma from autoimmune pancreatitis, with four studies reporting lower ADC values in AIP than in PDAC, but three reporting the opposite result [30] (Figure 16 and Figure 17).

DWI play a crucial role in the identification of worrisome feature in pancreatic cystic lesions and, according to some authors, high b-value DWI may help in the detection and classification of solid lesions in IPMN [31,32] (Figure 18), thus contributing to the differentiation between benign and malignant IPMNs [33]. On unenhanced MRI, DWI with MRCP could improve the diagnosis of malignant IPMN with a better prediction of invasive IPMN [34,35]. According to some authors, DWI can be useful in differentiating serous from mucinous cystic lesions; a threshold value of 3 × 10 − 3 mm^2^/sec on the ADC maps could help to distinguish mucinous from serous lesions (the latter with lower ADC values) with an accuracy rate of 77–81% and a good correspondence with anatomo-pathological outcomes [36,37] (Figure 19 and Figure 20). Finally, DWI is a non-invasive tool that allows the detection of infection in acute pancreatitis-associated collections; an infected collection shows restricted diffusion and ADC values in the central parts significantly different from non-infected groups [38]. Finally, DWI offers important information in the assessment of the onset of small pancreatic carcinoma by using short protocols [39].

**e. T1 mapping:** myocardial fibrosis and myocardial deposition disease were investigated with T1 mapping and, in the latter years, the availability of fast volumetric T1 mapping techniques has led to its application in other organs, such as the liver, with the aim to identify and quantify liver fibrosis [40]. Recently, an added value of T1 mapping is represented by the detection of early fibrotic changes in mild chronic pancreatitis. Thus, T1 mapping could be able to distinguish a normal pancreas from mild chronic pancreatitis, opening a possibility of CP early diagnosis [41].

**f. MR cholangiopancreatography (MRCP):** The same sequences used for T2-weighted biliary imaging can also be applied in the visualization of pancreatic ducts as well as cystic lesions of the pancreas. A better image quality can be reached after the oral administration of contrast material, which significantly reduces the signal of the fluid within the stomach and duodenum such as some fruit juices (pineapple, blueberry, cranberry, etc.) [42]. A superior image quality of MRCP can be assessed with a prior injection of paramagnetic contrast agent; the T2* effect of gadolinium suppresses the signal from the vessel and fluid within the interstitial compartment of the pancreas, sparing the signal from the pancreatic and bile ducts [43].

MRCP is part of a standard protocol of MRI of the pancreas thanks to its fast and accurate visualization of the pancreatic duct and its alteration. MRCP is an important imaging tool to identify alterations of the pancreatic duct, from benign (e.g., chronic pancreatitis) (Figure 15) to border line (IPMN) (Figure 18) to malignant, which can show the presence of a pancreatic carcinoma even before it is visible on other sequences (Figure 21), and it is useful in the evaluation of cystic tumors of the pancreas [44].

**g. Secretin MRCP (S-MRCP)** consists in the application of MRCP after the injection of secretin, a hormone which stimulates the exocrine pancreas to produce fluid and bicarbonate [45]. The advantages of secretin-enhanced MRCP are therefore both morphological and functional:

- **Morphological**: Offers better visualization of the MPD and easier establishment of the duct anatomical variants, such as the pancreas divisum, discerning stenosis, dilatation, obstruction and irregular borders of the duct. As a whole, S-MRCP increases the negative predictive value of MR imaging of the pancreas [46].

- **Functional**: A sign of early chronic pancreatitis is the visualization of side branches at body-tail after secretin injection; a hindered pancreatic juice outflow is related to a prolonged and abnormal dilatation of the MPD (>3 mm 10 min after secretin injection); the parenchymogram (parenchymal enhancement) is a sign of recurrent acute pancreatitis (Figure 22); a decrease in pancreatic exocrine reserve is explained by a reduced duodenal filling [47]. Secretin-stimulated MRCP images are improved in comparison to standard MRCPs in many aspects; moreover, S-MRCP could support diagnosis and clinical decision making, especially in patients with acute, acute recurrent or chronic pancreatitis, yielding a better identification of patients in need of therapeutic ERCP [48]. Finally, S-MRCP can help in differentiating a malignant from a benign focal stenosis of the wirsung duct [49].

In Table 2, different practical protocols of MR pancreatic imaging in different clinical situations are offered, together with the most useful indications of MRI in different clinical situations.

## 4. Conclusions

MR imaging of the liver and pancreas is a powerful tool which helps in managing complex cases. Its multiparametric approach, lack of ionizing radiation, high sensitivity to contrast media and availability of functional sequences able to give quantitative non-morphological information make MRI an important technique able to solve many problems.

The lack of ionizing radiations makes this technique an important alternative to CT in case of repeated follow-up, especially in children or young patients, while the multiparametric approach, either morphological or functional, make it the final non-invasive approach in cases where other imaging techniques such as US, CEUS and CT are not able to give useful information.

## Figures and Tables

**Figure 1 diagnostics-14-00693-f001:**
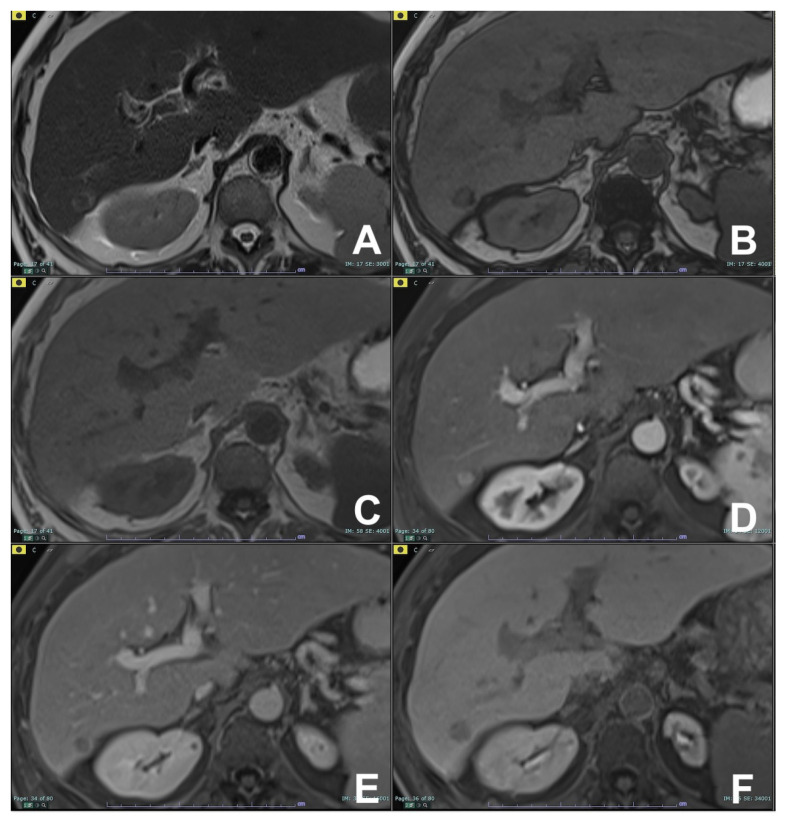
(**A**–**F**): HCC with fatty changes. On T2w (**A**), a lesion at segment VI can be appreciated. For out- (**B**) and in-phase (**C**) T1w, the lesion shows fatty content, visible with a loss of signal in out-of-phase. The lesion is hypervascular in the arterial phase (**D**) with washout in the venous phase (**E**). At the hepatobiliary phase after liver-specific MR contrast agent (Multihance, Bracco), the lesion is hypointense (**F**).

**Figure 2 diagnostics-14-00693-f002:**
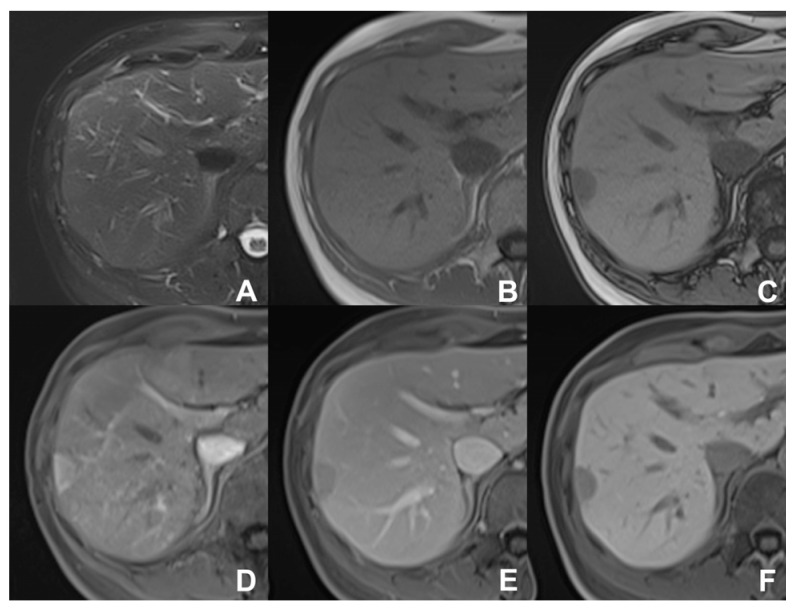
(**A**–**F**): Hepatic adenoma. On T2w (**A**) and T1w in-phase (**B**), no significant lesions are appreciable. On the T1w out-of-phase (**C**) image, a hypointense lesion is appreciable due to intracellular fat content. After injection of liver-specific MR contrast agent Multihance (Bracco, Milano), the lesion is hypervascular in the arterial phase (**D**),with washout in the venous phase (**E**) and no contrast uptake in the hepatobiliary phase (**F**).

**Figure 3 diagnostics-14-00693-f003:**
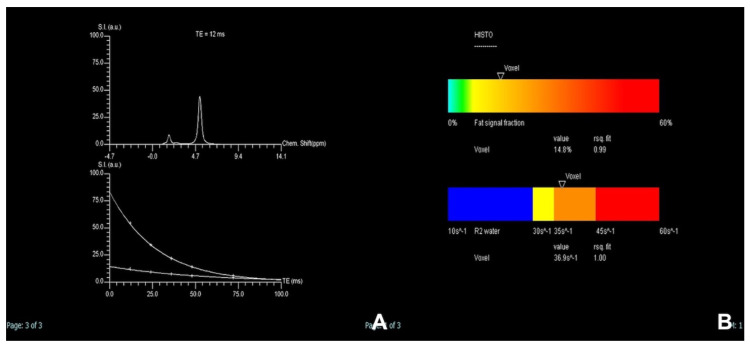
(**A**,**B**) Fat/Iron quantification. (**A**) Top: spectral separation of iron and fat. Bottom: signal decay in the different TEs. (**B**) Top: fat fraction (14.8%). Bottom: R2water (36.9 SEC -1), expression of iron deposit.

**Figure 4 diagnostics-14-00693-f004:**
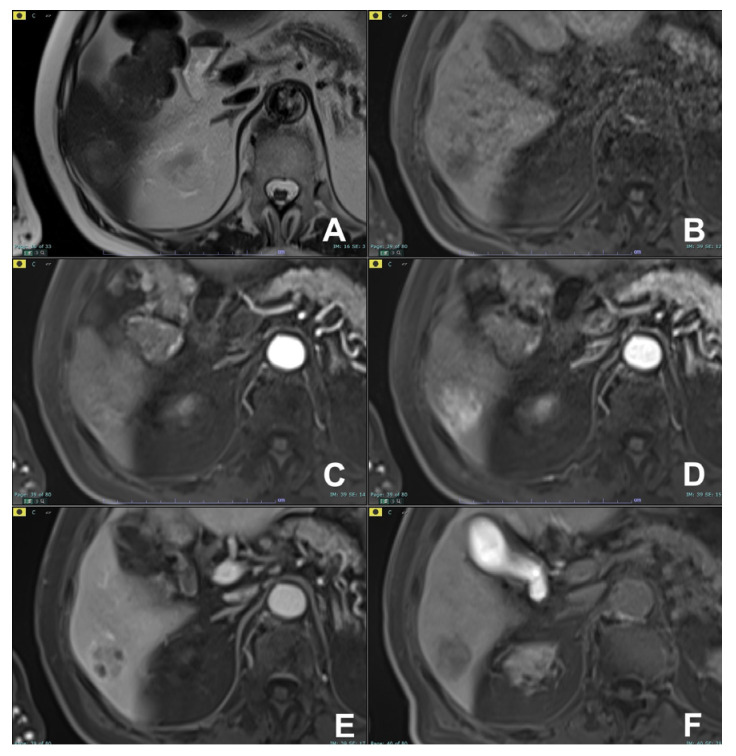
(**A**–**F**): HCC with multi-arterial phase. On T2w (**A**), a slight hyperintense lesion is appreciated at the VI segment, which appears hypointense on T1w (**B**). The lesion is hypervascular (**C**) and better appreciated in the second arterial acquisition (**D**) with washout at the venous phase (**E**). At the hepatobiliary phase after liver-specific MR contrast agent (Multihance, Bracco), the lesion is hypointense (**F**).

**Figure 5 diagnostics-14-00693-f005:**
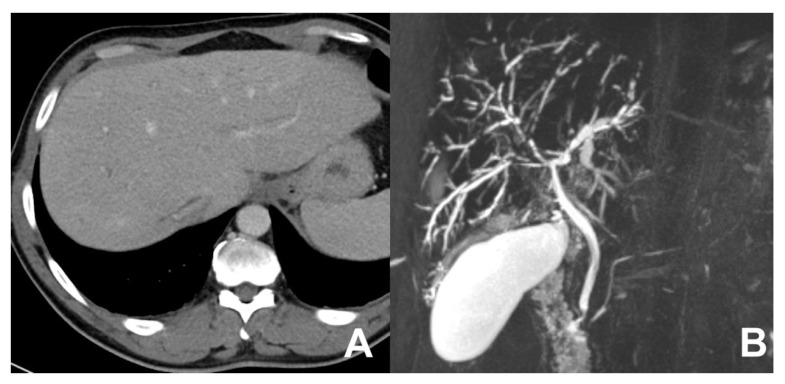
(**A**,**B**) PSC, comparison of CT and MRCP. With CT (**A**), only some slight ectasia of peripheral biliary ducts can be appreciated. With MRCP (**B**), the multiple stenosis of biliary ducts are clearly visible.

**Figure 6 diagnostics-14-00693-f006:**
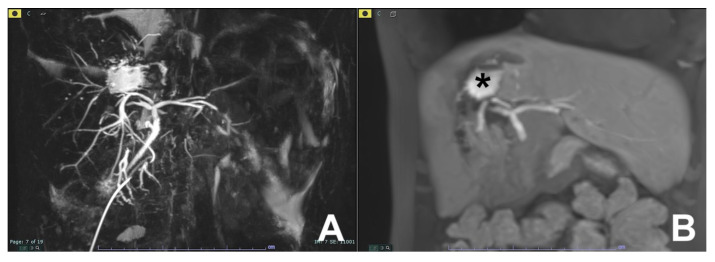
(**A**,**B**) Bile leak after surgical resection. At T2-MRCP (**A**), a fluid collection is visible in the dome of the liver. After injection of liver-specific MR contrast agent (Primovist, Bayer, Berlin, Germany) during the hepatobiliary phase (**B**), a collection of contrast media is appreciated at the same level of the fluid collection (*), indicating a bile leak.

**Figure 7 diagnostics-14-00693-f007:**
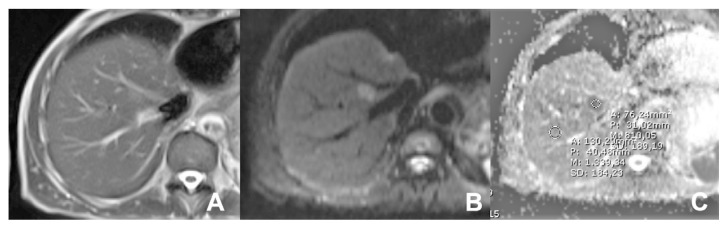
(**A**–**C**) Metastasis from melanoma. On T2w (**A**), a slight hyperintense lesion is poorly visible at the VII segment. On DWI b50 (**B**), the lesion is clearly visible. In the ADC map (**C**), the lesion appears hypointense due to its malignant nature.

**Figure 8 diagnostics-14-00693-f008:**
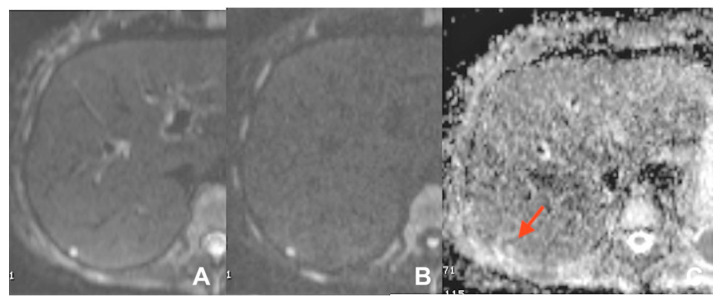
(**A**–**C**) Small hemangioma. On DWI b50 (**A**), the lesion is hyperintense, still maintaining the hyperintensity of DWI b800 (**B**) due to the “T2 shine-throw effect. On the ADC map (**C**), the lesion is hyperintense (arrow) due to unrestricted diffusion.

**Figure 9 diagnostics-14-00693-f009:**
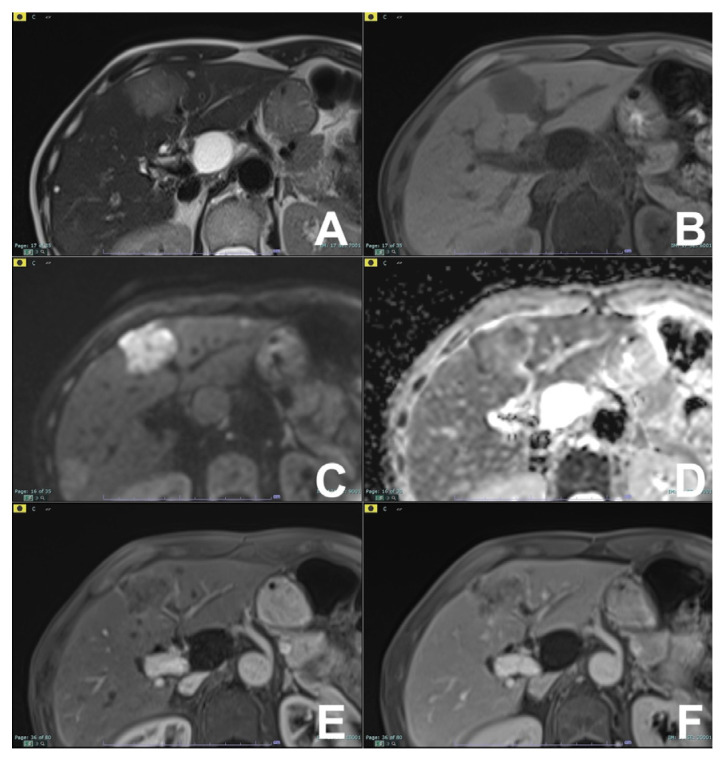
(**A**–**F**) Metastasis from colon carcinoma. On T2w (**A**), the lesion is slightly hyperintense and hypointense on T1w (**B**). On DWI b800 (**C**), the lesion is hyperintense but shows a central hyperintense area on the ADC map (**D**) due to necrotic changes, appreciable in the arterial (**E**) and venous (**F**) phases.

**Figure 10 diagnostics-14-00693-f010:**
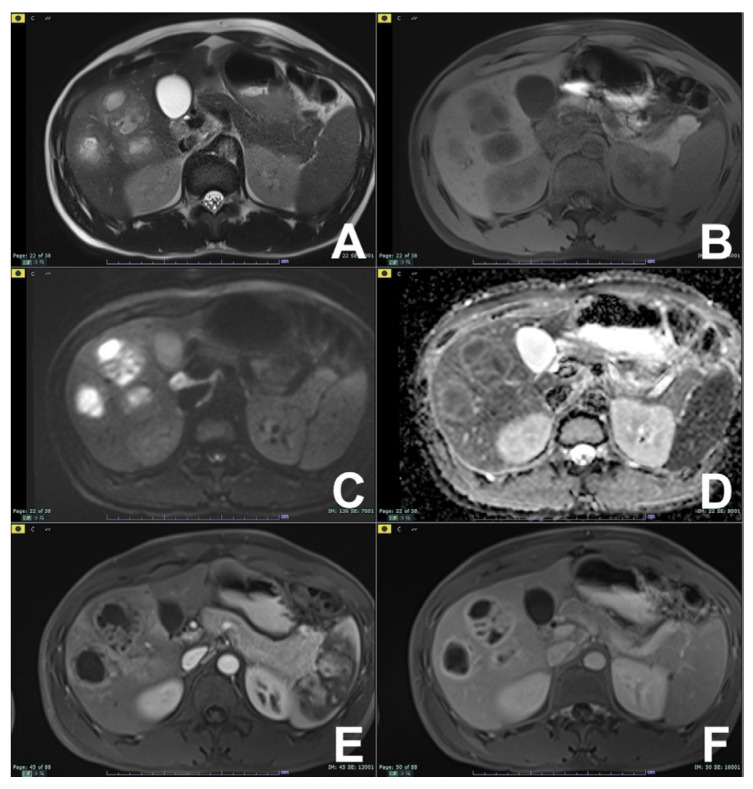
(**A**–**F**) Liver abscess. On T2w (**A**), the lesion is slightly hyperintense with some foci markedly hyperintense and hypointense on T1w (**B**). On DWI b800 (**C**), the lesion is hyperintense but shows a central hypointense area on the ADC map (**D**) due to coagulative necrosis, appreciable on the arterial (**E**) and venous (**F**) phases.

**Figure 11 diagnostics-14-00693-f011:**
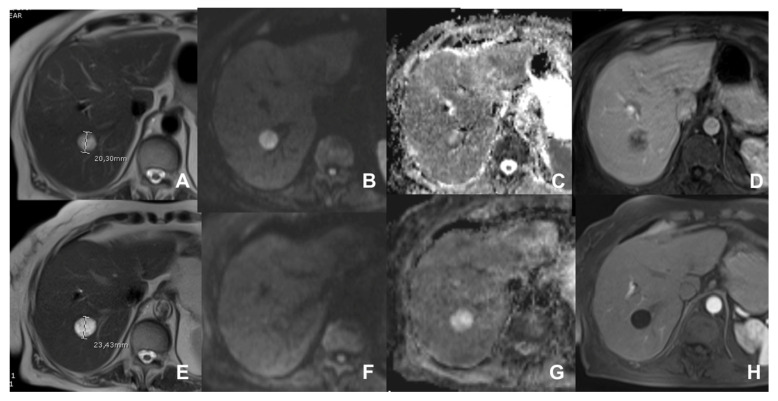
(**A**–**H**) Metastasis from GIST before (**A**–**D**) and after (**E**–**H**) therapy. Before therapy, the lesion shows cystic changes (**A**) but with areas of restricted diffusion (**B**,**C**) and with internal enhancement (**D**). After therapy, the lesion is enlarged (**E**) but with no restricted diffusion (**F**,**G**) and no enhancement (**H**).

**Figure 12 diagnostics-14-00693-f012:**
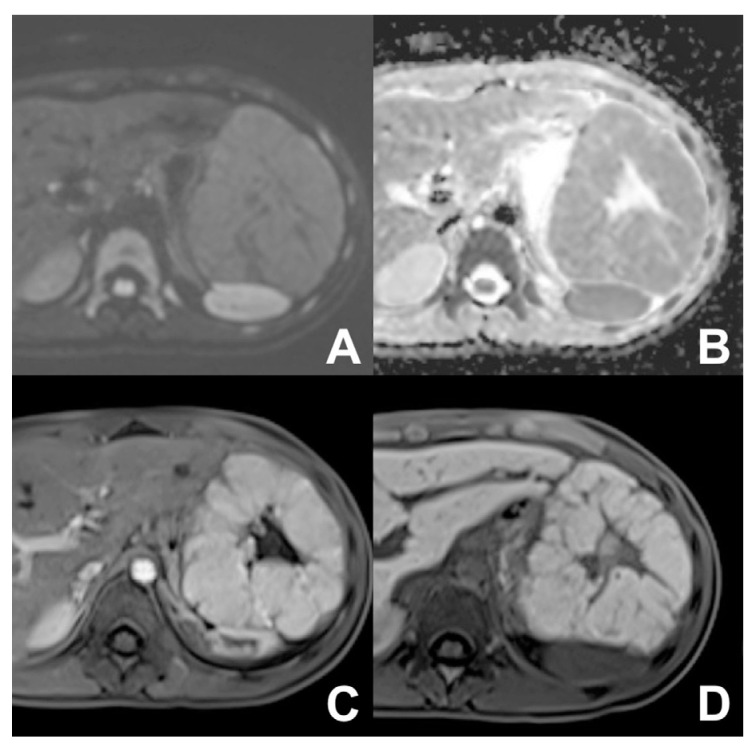
(**A**–**D**) FNH. A large lesion with a central scar and no significant restriction (**A**,**B**) is appreciable at the left lobe. After injection of liver-specific MR contrast agent Multihance (Bracco, Milano), the lesion is hypervascular (**C**) with uptake of contrast agent in the hepatobiliary phase (**D**).

**Figure 13 diagnostics-14-00693-f013:**
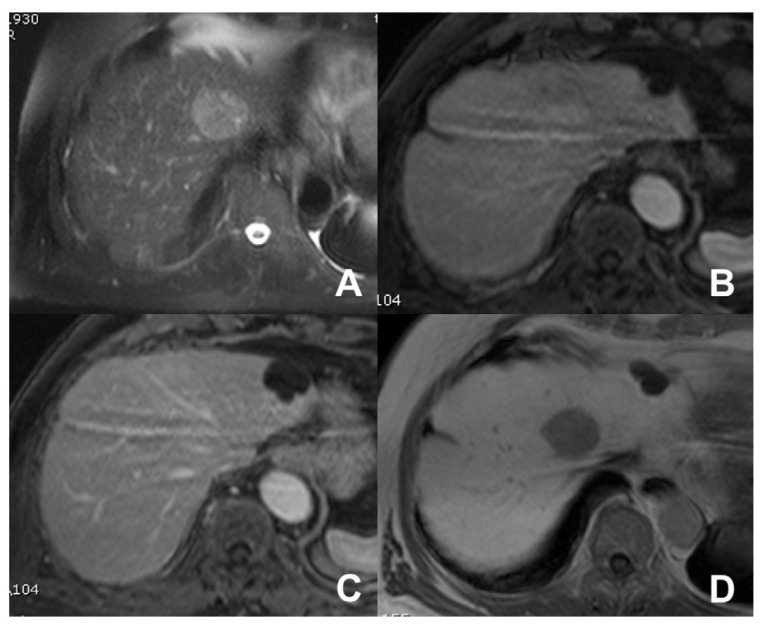
(**A**–**D**) HCC, well differentiated. On T2w (**A**), a round slightly hyperintense lesion is appreciable in the left lobe. After injection of liver-specific MR contrast agent Multihance (Bracco, Milano), the lesion shows a slight enhancement in the arterial phase (**B**) with no washout in the venous phase (**C**) but is markedly hypointense in the hepatobiliary phase (**D**).

**Figure 14 diagnostics-14-00693-f014:**
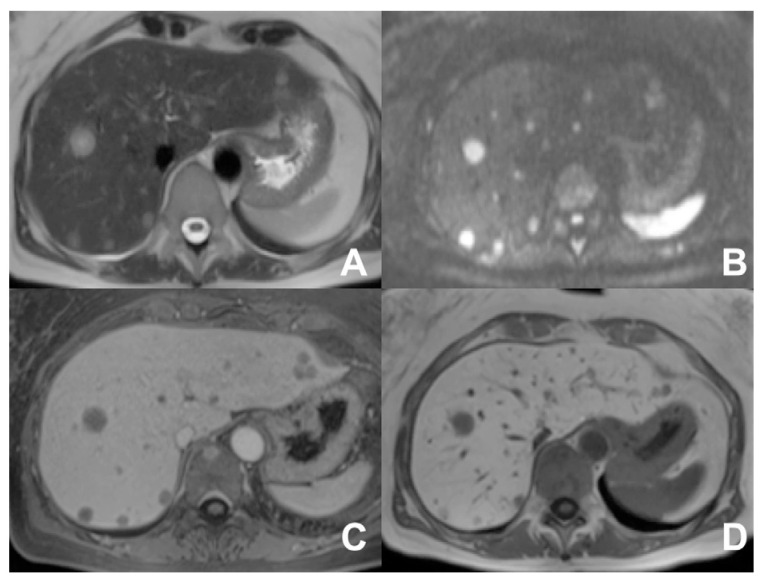
(**A**–**D**) Metastases from breast carcinoma. On T2w (**A**), a large hyperintense lesion is appreciable in the right lobe as well as other small slightly hyperintense lesions better appreciable on DWI b800 (**B**). After injection of liver-specific MR contrast agent Primovist (Bayer, Berlin), the lesions are hypovascular (**C**) with no contrast uptake in the hepatobiliary phase (**D**).

**Figure 15 diagnostics-14-00693-f015:**
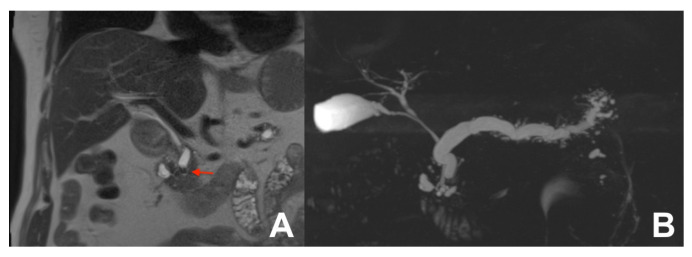
(**A**,**B**) Chronic calcifying pancreatitis. On HASTE T2w coronal (**A**), a small defect in the prepapillary wirsung duct (arrow) due to a calculi. On MRCP (**B**), a diffuse dilatation of the wirsung duct with side branch ectasia is clearly appreciable.

**Figure 16 diagnostics-14-00693-f016:**
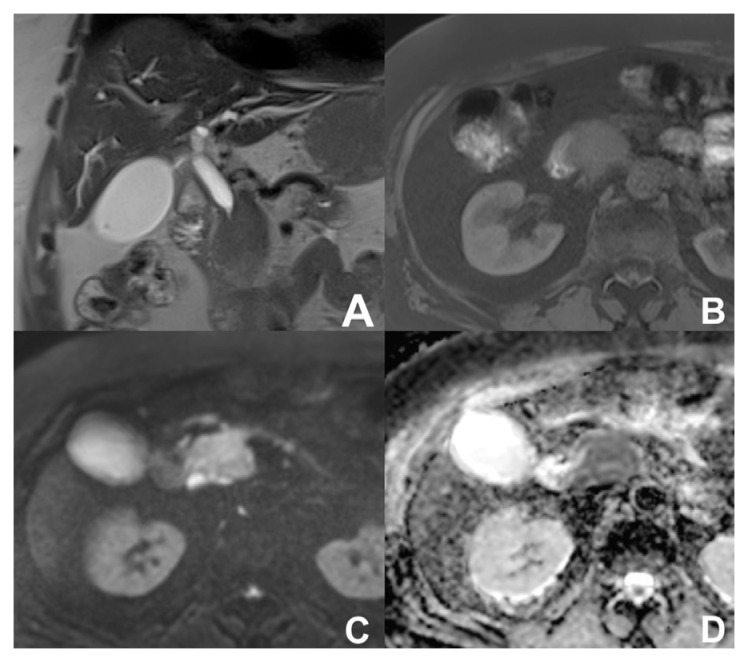
(**A**–**D**) Focal autoimmune pancreatitis. On HASTE T2w coronal (**A**), a large mass in the head of the pancreas with stenosis of the choledochus is appreciable. The mass is hypointense on T1w fat sat (**B**) and shows marked restricted diffusion on DWI (**C**) and the ADC map (**D**).

**Figure 17 diagnostics-14-00693-f017:**
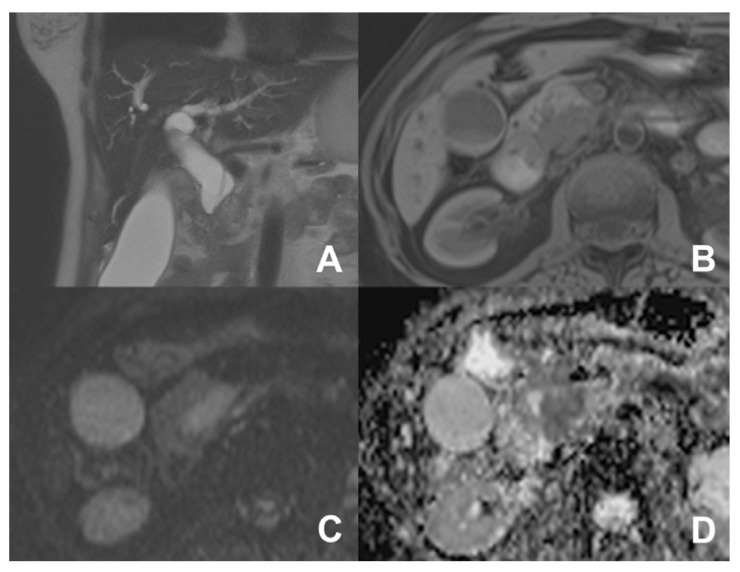
(**A**–**D**) Pancreatic cancer. On HASTE T2w coronal (**A**), a large mass in the head of the pancreas with marked stenosis of the choledochus is appreciable. The mass is hypointense on T1w fat sat (**B**) and shows restricted diffusion on DWI (**C**) and on the ADC map (**D**).

**Figure 18 diagnostics-14-00693-f018:**
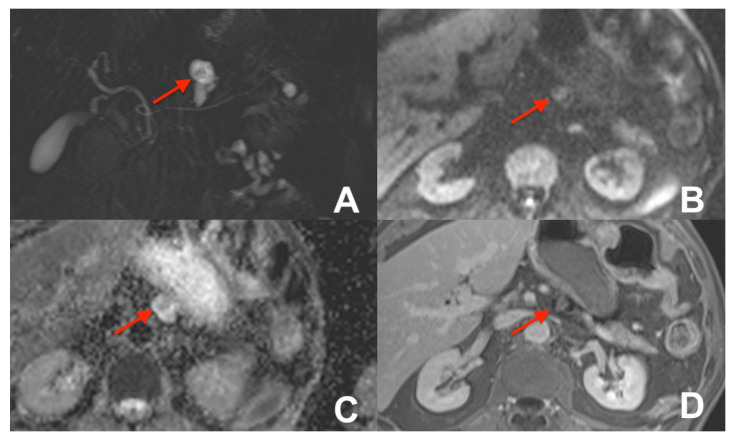
(**A**–**D**) BD-IPMN. On MRCP (**A**), a branch-duct IPMN with a small defect (arrow) is appreciable, which shows restricted diffusion at DWI (**C**) and on the ADC map (**C**) and enhancement after MR contrast media injection (**D**).

**Figure 19 diagnostics-14-00693-f019:**
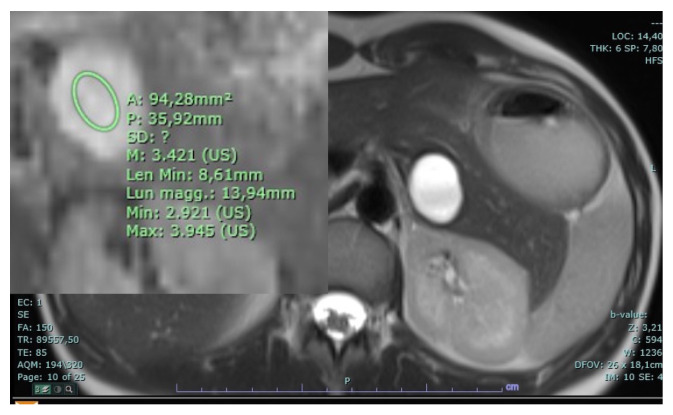
Mucinous cystoadenoma. A large cystic lesion with smooth walls is appreciable in the tail of the pancreas with an ADC value above three.

**Figure 20 diagnostics-14-00693-f020:**
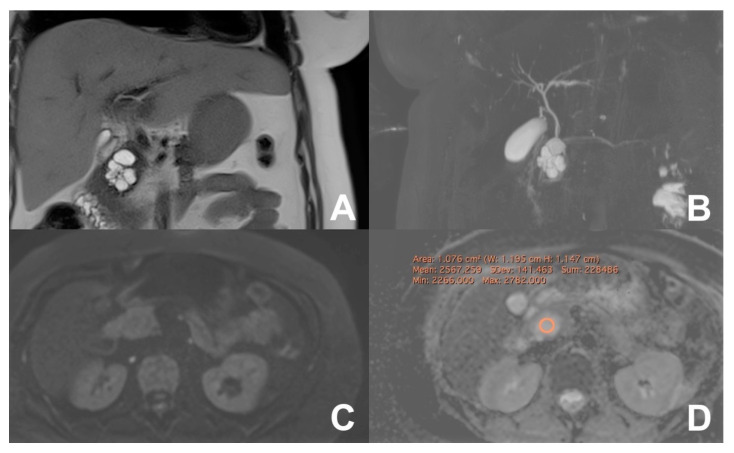
(**A**–**D**). Serous cystoadenoma. On HASTE T2w coronal (**A**), a cystic lesion with lobulated margins is appreciable in the head of the pancreas, better appreciable on MRCP (**B**). On DWI b800, no restriction is appreciable and at the ADC map the value is 2.5.

**Figure 21 diagnostics-14-00693-f021:**
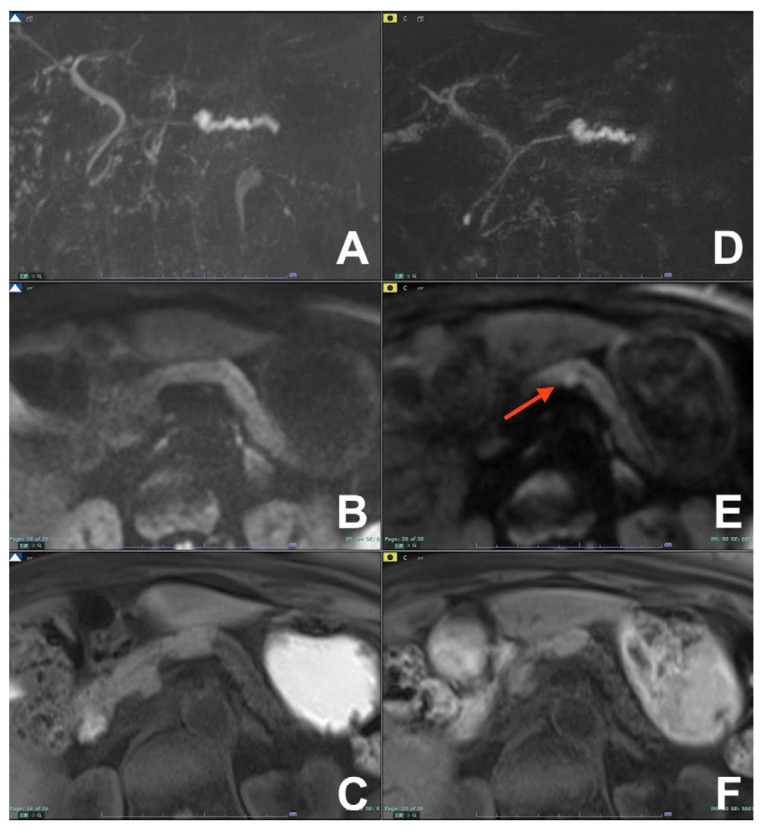
(**A**–**F**) Early detection of pancreatic cancer. (**A**–**C**): a stenosis of the wirsung duct is appreciable with no evidence of mass. (**D**–**F**): after 4 months a small hyperintense foci at the level of the stenosis is appreciable at DWI b800 (red arrow) (**E**) with no other evidence of mass on T1w (**F**).

**Figure 22 diagnostics-14-00693-f022:**
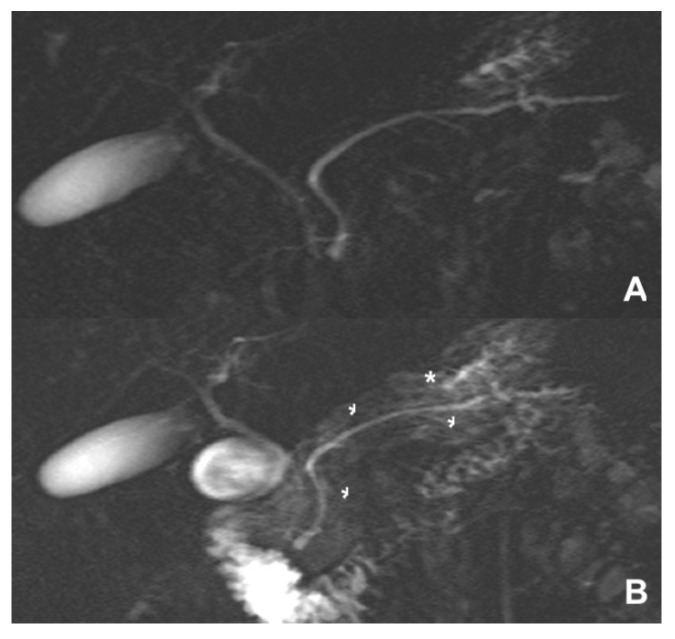
(**A**,**B**) Sphincter of Oddi dysfunction. (**A**) MRCP: no significant alterations are visible. (**B**) After secretin injection, diffuse parenchymal enhancement (*) of the pancreas is visible.

**Table 1 diagnostics-14-00693-t001:** Practical hepatic MR protocols in different clinical indications with indication of the usefulness of MRI.

Clinical Indication	Sequence	Plane	CM Delay	HB Phase (Only after LS-MRCA)	Note
**Healthy liver: characterization of FLL**	HASTE T2	AXIAL	--	--	Anatomy and liquids analysis
INDICATIONS: MRI is the technique of choice in young patients and pregnant women after a unclear US finding. Moreover, it is useful even after a CT with undefined diagnosis for all patients.	HASTE T2	CORONAL	--	--	Anatomy and liquids analysis
	DWI b 0–50–400–800	AXIAL	--	--	Restricted diffusion most likely malignant
	GRE T1 IN/OUT	AXIAL	--	--	Steatosis
	GRE T1 3D DYNAMIC	AXIAL	PRE-ART 25″-PORTAL 70″-LATE 180″	YES (if the lesion is hypervascular)	Benign hypervascular hepatocitic lesions appear hyperintense in HBP

**Cirrhotic liver: characterization of FLL/Follow-up after treatment**	HASTE T2	AXIAL	--	--	Anatomy and liquids analysis
INDICATIONS: MRI is the technique of choice in young patients and pregnant women after a unclear US finding. Moreover, it is useful even after a CT with undefined diagnosis for all patients.	HASTE T2	CORONAL	--	--	Anatomy and liquids analysis
	DWI b 0–50–400–800	AXIAL	--	--	High signal in b 800 suspicious for HCC
	GRE T1 IN/OUT	AXIAL	--	--	Steatosis
	GRE T1 3D DYNAMIC	AXIAL	PRE-ART 25″-PORTAL 70″-LATE 180″	YES (if the lesion shows atypical enhancement)	Hypointensity in HBP suspicious for HCC

**Follow-up oncological Patient**	HASTE T2	AXIAL	--	--	Anatomy and liquids analysis
INDICATIONS: MRI is the technique of choice in young patients and pregnant women after a unclear US finding. Moreover, it is useful even after a CT with undefined diagnosis for all patients. It can be alternated to CT in young patients with a long follow-up.	HASTE T2	CORONAL	--	--	Anatomy and liquids analysis
	DWI b 0–50–400–800	AXIAL	--	--	DWI b50 increases the sensitivity of mets detection
	GRE T1 IN/OUT	AXIAL	--	--	Steatosis
	GRE T1 3D DYNAMIC	AXIAL	PRE-ART 25″-PORTAL 70″-LATE 180″	YES	HBP increases the sensitivity of metastases detection

**Liver abscess/Biliary inflammation/Follow-up in biliary diseases (e.g., PSC)/cholangiocarcinoma**	HASTE T2	AXIAL	--	--	Anatomy and liquids analysis
INDICATIONS: MRI is the technique of choice in follow-up of PSC or secondary sclerosing cholangitis. Useful for dd between abscess and necrotic malignant lesions. In case of hilar cholangiocarcinoma, it is useful to assess the involvement of biliary ducts as well as the evaluation of biliary anatomy and variants.	HASTE T2	CORONAL	--	--	Anatomy and liquids analysis
	DWI b 0–50–400–800	AXIAL	--	--	Abscess in ADC map is hypointense
	GRE T1 IN/OUT	AXIAL	--	--	Steatosis
	GRE T1 3D DYNAMIC	AXIAL	PRE-ART 25″-PORTAL 70″-LATE 180″	NO	For better characterization of abscess and cholangitis
	MRCP 3D	OBLIQUE CORONAL			Biliary anatomy and calculi or stenosis

**Biliary calculi**	HASTE T2	AXIAL	--	--	Anatomy and liquids analysis
Indications: MRI is the technique of choice to detect biliary calculi either in the hepatic parenchyma or in the choledocus. Useful to confirm/exclude biliary calculi in patients with acute pancreatitis.	HASTE T2	CORONAL	--	--	Anatomy and liquids analysis
	GRE T1 IN/OUT	AXIAL	--	--	Steatosis
	MRCP 3D	OBLIQUE CORONAL	--	--	use MIP and sub MIP for better detection of calculi

**Biliary leak after surgery or trauma**	HASTE T2	AXIAL	--	--	Anatomy and liquids analysis
INDICATIONS: MRI, thanks to the use of liver-specific MR contrast agents, can easily detect the site of bile leak.	HASTE T2	CORONAL	--	--	Anatomy and liquids analysis
	MRCP 3D	OBLIQUE CORONAL	--	--	Biliary anatomy and calculi
	GRE T1 IN/OUT	AXIAL	--	--	Steatosis
	GRE T1 3D DYNAMIC	AXIAL	Not necessary	YES high resolution	Useful for leak

**Iron/fat quantification**	HASTE T2	AXIAL	--	--	Anatomy and liquids analysis
Indications: MRI is the technique of choice to quantify the amount of fat or iron overload.	HASTE T2	CORONAL	--	--	Anatomy and liquids analysis
	GRE T1 IN/OUT	AXIAL	--	--	Steatosis
	GRE Multi echo	AXIAL	--	--	For the quantification of fat and iron content


**Table 2 diagnostics-14-00693-t002:** Practical pancreatic MR protocols in different clinical indications with indication of the usefulness of MRI.

Clinical Indication	Sequence	Plane	CM Delay	Note
**Acute pancreatitis (AP)**	HASTE T2	AXIAL	--	Anatomy and analysis of the content of collections
Indications: MRI is the technique of choice to detect biliary calculi either in the hepatic parenchyma or in the choledocus. It is useful to confirm/exclude biliary calculi in patients with AP. Moreover, MRI is able to better characterize the content of collections, thus allowing an appropriate management, either percutaneous/endoscopic or surgical. MRI can be used to follow-up AP in young and child-bearing patients. Finally, with DWI it is possible to better identify an infected collection.	HASTE T2	CORONAL	--	Anatomy and analysis of the content of collections
	T1 GRE FS	AXIAL	--	Pancreatic parenchima assessment
	DWI b 0–50–400–800	AXIAL	--	Infected collections appear hypointense in the ADC map
	GRE T1 3D DYNAMIC	AXIAL	Pre- 25″–70″–180″	Not always necessary
	MRCP 3D/2D	OBLIQUE CORONAL		Anatomy of wirsung duct

**Recurrent Acute pancreatitis (RAR)**	HASTE T2	AXIAL	--	Anatomy and liquids analysis
Indications: MRI is the technique of choice to detect pancreatic abnormalities which can cause episodes of recurrent pancreatitis. Moreover, the use of secretin is able to provide functional information useful for the identification of the causes of RAR not otherwise available.	HASTE T2	CORONAL	--	Anatomy and liquids analysis
	GRE T1 FS	AXIAL	--	Pancreatic parenchima assessment
	DWI b 0–50–400–800	AXIAL	--	Not necessary if pancreatic parenchima is normal at GRE T1 FS
	MRCP 3D/2D	OBLIQUE CORONAL		Anatomy of wirsung duct
	MRCP 2D with secretin	OBLIQUE CORONAL		Functional information

**Chronic pancreatitis (CP)**	HASTE T2	AXIAL	--	Anatomy and liquids analysis
Indications: MRI is complementary to CT in the diagnosis and management of CP. Although it is not able to visualize calcifications, with MRCP it is able to detect early changes of CP, especially after secretin. Lack of ionizing radiation makes MRI the technique of choice in cases with long follow-up.	HASTE T2	CORONAL	--	Anatomy and liquids analysis
	GRE T1 FS	AXIAL	--	Pancreatic parenchima assessment
	DWI b 0–50–400–800	AXIAL	--	Restriction within pancreas can suggest an abnormal condition
	GRE T1 3D DYNAMIC	AXIAL	Pre- 25″–70″–180″	In case of suspicious mass
	MRCP 3D/2D	OBLIQUE CORONAL		Anatomy of wirsung duct
	MRCP 2D with secretin	OBLIQUE CORONAL		For the early diagnosis of CP

**Differential diagnosis pancreatic cancer (PC) from mass forming pancreatitis (e.g., paraduodenal pancreatitis-PDP, autoimmune pancreatitis-AIP)**	HASTE T2	AXIAL	--	Anatomy and liquids analysis
Indications: The use of multiparametric imaging (DWI, contrast enhanced) allows one to differentiate with substantial accuracy a mass forming pancreatis from a pancreatic carcinoma.	HASTE T2	CORONAL	--	Anatomy and liquids analysis
	GRE T1 FS	AXIAL	--	Pancreatic parenchima assessment
	DWI b 0–50–400–800	AXIAL	--	Restricted diffusion most likely to be PC or AIP
	GRE T1 3D DYNAMIC	AXIAL	Pre- 25″–70″–180″	PC is most likely hypovascular; mass forming pancreatitis usually shows delayed homogeneous enhancement
	MRCP 3D/2D	OBLIQUE CORONAL		Anatomy of wirsung duct
	MRCP 2D with secretin	OBLIQUE CORONAL		For the differential diagnosis of pancreatic duct stenosis (“duct penetrating sign”)

**Neuroendocrine tumor of the pancreas (PNEN)**	HASTE T2	AXIAL	--	Anatomy and liquids analysis
Indications: MRI is useful for the detection of PNENs thanks to DWI and contrast enhancement. DWI can suggest the degree of differentiation of PNENs as high grade PNENS (G2-G3) usually show a marked restriction of DWI. Useful for the follow-up of small low-grade PNENs which cannot be resected.	HASTE T2	CORONAL	--	Anatomy and liquids analysis
	GRE T1 FS	AXIAL	--	pancreatic parenchima assessment
	DWI b 0–50–400–800	AXIAL	--	Restricted diffusion tipical of PNEN. Useful for multifocal PNENs
	GRE T1 3D DYNAMIC	AXIAL	Pre- 25″–70″–180″	PNEN is most likely hypervascular
	MRCP 3D/2D	OBLIQUE CORONAL		Anatomy of wirsung duct

**Cystic lesion of the pancreas (CPL)**	HASTE T2	AXIAL	--	Anatomy and cysts morphology
Indications: MRI is the technique of choice in the diagnosis and management of CPL. T2, MRCP, DWI and contrast enhancement allow one to differentiate the different lesions and suggest an optimal management. As a rule of thumb, benign cystic lesions (serous cystosadenoma) do not require surgical resection or follow-up; border line or malignant lesions (mucinous cystoadenoma; main duct IPMN, BD-IPM with high-risk stigmata) require surgical resection if the patient is fit for surgery, while low-risk cystic lesions (BD-IPMN with no worrisome or high-risk stigmata) require follow-up (see below).	HASTE T2	CORONAL	--	Anatomy and cysts morphology
	GRE T1 FS	AXIAL	--	Pancreatic parenchima assessment
	DWI b 0–50–400–800	AXIAL	--	Restricted diffusion inside the CPL can be considered a worrisome feature and injection of contrast is indicated
	GRE T1 3D DYNAMIC	AXIAL	Pre- 25″–70″–180″	Useful for the characterization of worrisome features (thick walls, septa, nodules, restricted diffusion)
	MRCP 3D/2D	OBLIQUE CORONAL		Relationship between the wirsung duct and the cystic lesion

**Follow-up of IPMN**	HASTE T2	AXIAL	--	Anatomy and cysts morphology
Indications: in case of BD-IPMN with no worrisome or high risk stigmata.	HASTE T2	CORONAL	--	Anatomy and cysts morphology
	GRE T1 FS	AXIAL	--	Pancreatic parenchima assessment
	DWI b 0–50–400–800	AXIAL	--	Restriction inside the IPMN can be considered a worrisome feature, and injection of contrast is indicated
	GRE T1 3D DYNAMIC	AXIAL	Pre- 25″–70″–180″	In case of worrisome features (thick walls, septa, nodules, restricted diffusion inside the cyst), otherwise not necessary
	MRCP 3D/2D	OBLIQUE CORONAL		Comprehensive evaluation of the wirsung duct and cystic lesions

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
