# Peer review of "Imaging of the Liver and Pancreas: The Added Value of MRI"

_diagnostics, 2024, doi:10.3390/diagnostics14070693_

Round 1

Reviewer 1 Report

Comments and Suggestions for Authors

Dear Authors,

There are numerous strengths in this study including current and novel MRI application in liver and pancreas imaging. I found this study makes a valuable contribution to the literature.

I would add a table with the “perfect” protocol for the main pathologies, including MRI sequences, planes and contrast phase acquisition and delays. It’s a suggestion and not necessary but because of the great educational value of the paper, it would be useful for young radiologists. 

Author Response

Reviewer 1:

Dear Authors,

There are numerous strengths in this study including current and novel MRI application in liver and pancreas imaging. I found this study makes a valuable contribution to the literature.

I would add a table with the “perfect” protocol for the main pathologies, including MRI sequences, planes and contrast phase acquisition and delays. It’s a suggestion and not necessary but because of the great educational value of the paper, it would be useful for young radiologists. 

Answer: thank you for the comments. We added tables with the most practical MR protocols in the most frequent clinical situations

Reviewer 2 Report

Comments and Suggestions for Authors

The authors try to shed light on the value of MRI in both hepatic and pancreatic diseases with the advent of new contrast agents. Recently, with the advent of new imaging techniques, it is of utmost importance for clinicians to choose the right diagnostic imaging modality. The new data about the modern imaging modalities should be available to clinician, so, they can help their patients. 

I have few comments:

It is a very specified manuscript in radiology. If you can add a paragraph to explain the idea of MRI and how you can interpret it in a simplified way for clinicians.

Figure 15 footnote: prepapillar------> prepapillary

Page 4, Liver-specific MR contrast agents: e.v----> IV.

Can you suggest an algorithm for the diagnosis of hepatic focal lesions using MRI? 

Can you document a schedule for the MRI picture of different hepatic lesions?

For pancreatic lesions, it is well-known that endoscopic ultrasonography (EUS) is the best procedure used in both diagnosis and treatment of pancreatic lesions. You need to compare the EUS with MRI.

Author Response

Reviewer 2:

The authors try to shed light on the value of MRI in both hepatic and pancreatic diseases with the advent of new contrast agents. Recently, with the advent of new imaging techniques, it is of utmost importance for clinicians to choose the right diagnostic imaging modality. The new data about the modern imaging modalities should be available to clinician, so, they can help their patients. 

I have few comments:

  1. Figure 15 footnote: prepapillar------> prepapillary

Answer: done

  1. Page 4, Liver-specific MR contrast agents: e.v----> IV.

Answer: done

  1. It is a very specified manuscript in radiology. If you can add a paragraph to explain the idea of MRI and how you can interpret it in a simplified way for clinicians.
  2. Can you suggest an algorithm for the diagnosis of hepatic focal lesions using MRI? 
  3. Can you document a schedule for the MRI picture of different hepatic lesions?

Answer: I give a collective answer to questions 3-4-5.

We have already filled all the space available for a review paper in term of characters and images. Such a project (technique of MRI, illustrated cases of most frequent hepatic and pancreatic images, diagnostic algorithm) would require additional space not available (at least 30-40 images more and other 15 pages).

We added a short indication of MRI in different clinical situations in tabs 1 and 2.

  1. For pancreatic lesions, it is well-known that endoscopic ultrasonography (EUS) is the best procedure used in both diagnosis and treatment of pancreatic lesions. You need to compare the EUS with MRI.

Answer:  thank you for the comments. We corrected the errors and we added a comment of EUS at the beginning of pancreatic section.

Round 2

Reviewer 2 Report

Comments and Suggestions for Authors

It is OK although I hoped to see a diagnostic diagrammatic algorithm for the use of MRI in different hepatic and pancreatic lesions.